 **eLIFE**

# Evolution of insect olfactory receptors

**Christine Missbach[1]\*, Hany KM Dweck[1], Heiko Vogel[2], Andreas Vilcinskas[3], Marcus C Stensmyr[4], Bill S Hansson[1†], Ewald Grosse-Wilde[1]\*†**

[1]Department of Evolutionary Neuroethology, Max Planck Institute for Chemical Ecology, Jena, Germany; [2]Department of Entomology, Max Planck Institute for Chemical Ecology, Jena, Germany; [3]Institute of Phytopathology and Applied Zoology, Justus-Liebig-Universität Gießen, Gießen, Germany; [4]Department of Biology, Lund University, Lund, Sweden

**Abstract** The olfactory sense detects a plethora of behaviorally relevant odor molecules; gene families involved in olfaction exhibit high diversity in different animal phyla. Insects detect volatile molecules using olfactory (OR) or ionotropic receptors (IR) and in some cases gustatory receptors (GRs). While IRs are expressed in olfactory organs across Protostomia, ORs have been hypothesized to be an adaptation to a terrestrial insect lifestyle. We investigated the olfactory system of the primary wingless bristletail *Lepismachilis y-signata* (Archaeognatha), the firebrat *Thermobia domestica* (Zygentoma) and the neopteran leaf insect *Phyllium siccifolium* (Phasmatodea). ORs and the olfactory coreceptor (Orco) are with very high probability lacking in *Lepismachilis*; in *Thermobia* we have identified three Orco candidates, and in *Phyllium* a fully developed OR/Orco-based system. We suggest that ORs did not arise as an adaptation to a terrestrial lifestyle, but evolved later in insect evolution, with Orco being present before the appearance of ORs.

**\*For correspondence:**
cmissbach@ice.mpg.de (CM);
Grosse-Wilde@ice.mpg.de (EG-W)

†These authors contributed equally to this work as senior authors.

**Competing interests:** The authors declare that no competing interests exist.

**Reviewing editor**: Gáspár Jékély, Max Planck Institute for Developmental Biology, Germany

## Introduction

All living organisms, including bacteria, protozoans, fungi, plants, and animals, detect chemicals in their environment. The sensitivity and chemical range of animal olfactory systems is remarkable, enabling animals to detect and discriminate between thousands of different odor molecules. Although there is a striking evolutionary convergence towards a conserved organization of signaling pathways in vertebrate and invertebrate olfactory systems (*Hildebrand and Shepherd, 1997*), the involved receptor gene families evolved independently. The molecular identity of olfactory receptors was first unraveled in vertebrates (*Buck and Axel, 1991*). In mammals, as many as 1000 heterotrimeric GTP-binding protein (or G protein)-coupled receptors are considered to be employed in olfactory discrimination (*Buck and Axel, 1991*). A similar number of chemoreceptors, with about 1300 receptor genes and 400 pseudogenes, have been hypothesized for *Caenorhabditis elegans* (*Robertson and Thomas, 2006*).

All data on insect olfactory receptors are based on studies investigating the neopteran insects (overview of insect order relationship is given in *Figure 1*). The identity of receptors involved in olfaction in the evolutionarily more ancient apterygote insects (Archaeognatha, Zygentoma) and paleopteran insects (Odonata and Ephemeroptera) is thus completely unknown. In neopteran insects (Polyneoptera, Paraneoptera, and Holometabola) most volatile stimuli are recognized by members of the olfactory receptor family (ORs). ORs are multitransmembrane domain proteins unrelated to nematode or vertebrate olfactory receptors (*Mombaerts, 1999*; *Robertson, 2001*; *Hill et al., 2002*), displaying a distinct membrane topology (*Benton et al., 2006*; *Lundin et al., 2007*). The number of functional OR genes varies from 10 in the human body louse *Pediculus humanus humanus* (*Kirkness et al., 2010*) to about 60 in *Drosophila melanogaster* (*Clyne et al., 1999*; *Gao and Chess, 1999*; *Vosshall et al., 1999*) and up to 350 OR genes in ants (*Zhou et al., 2012*). ORs have been suggested

**eLife digest** Detecting chemical cues can be a matter of life or death for insects, and many employ three families of receptor proteins to detect a broad range of odors. Members of one of these receptor families, the olfactory receptors, form a complex with another protein, the olfactory coreceptor that is essential for both positioning and stabilizing the receptor, as well as the actual function.

Crustaceans share a common ancestor with insects, and since they do not have olfactory receptors it has been proposed that these receptors evolved when prehistoric insects moved from the sea to live on land. According to this idea, olfactory receptors evolved because these ancestors needed to be able to detect odor molecules floating in the air rather than dissolved in water.

Previous research on insect olfactory receptors has focused on insects with wings. Missbach et al. have now used a wide range of techniques to investigate how evolutionarily older wingless insect groups detect scents. As all investigated groups evolved from a common ancestor at different times these experiments allow tracking of the historical development of olfactory receptors.

In the wingless species that is more closely related to the flying insects there was evidence of the presence of multiple coreceptors but not the olfactory receptors themselves. In the most basal insects no evidence for any part of the olfactory receptor-based system was found. This indicates that the main olfactory receptors evolved independently of the coreceptor long after the migration of insects from water to land. Missbach et al. suggest that olfactory receptors instead developed far later, around the time when vascular plants spread and insects developed the ability to fly.

to be distantly related to the gustatory receptors of arthropods, with some proteins containing a signature motif in the carboxyl terminus (*Scott et al., 2001*).

Insect olfactory receptors function as heteromultimers composed of at least one ligand-specific OR and the coreceptor Orco (*Vosshall et al., 1999*; *Elmore et al., 2003*; *Krieger et al., 2003*; *Larsson et al., 2004*; *Sato et al., 2008*; *Wicher et al., 2008*). Interestingly, while Orco (*Vosshall and Hansson, 2011*) is highly conserved among insects, the sequences of other olfactory receptor genes exhibit very little sequence similarity even within the same insect order (*Krieger et al., 2003*), complicating their identification. So far, Orco homologues have been identified in Lepidoptera, Diptera, Coleoptera, Hymenoptera, Hemiptera (*Krieger et al., 2003*; *Pitts et al., 2004*; *Smadja et al., 2009*), and Orthoptera (*Yang et al., 2012*). Neither Orco nor ORs are present in the genome of the crustacean *Daphnia pulex*, indicating that ORs are insect specific. However, GRs were found in Crustacea, just as in insects (*Peñalva-Arana et al., 2009*).

A second receptor family, the variant ionotropic glutamate receptors (IRs), is also involved in insect chemosensation (*Benton et al., 2009*). IRs act in combinations of up to three subunits; individual odor-specific receptors and one or two of the broadly expressed coreceptors IR25a, IR8a, and IR76b (*Abuin et al., 2011*). IRs are present in olfactory tissues across the Protostomia (*Croset et al., 2010*), for example two conserved members of this group were described in the *Daphnia* genome (*Croset et al., 2010*) and the coreceptor IR25a homologue is expressed in many, if not all mature OSNs of the American lobster *Homarus americanus* (*Hollins et al., 2003*) and the spiny lobster *Panulirus argus* (*Tadesse et al., 2011*). Since crustaceans are the closest relatives of insects (*Friedrich and Tautz, 1995*; *Boore et al., 1998*; *Regier et al., 2010*), IRs are most likely the ancient type of insect olfactory receptor.

But when and why did insect ORs evolve? Hexapods derived from an aquatic crustacean ancestor, probably in the Early Ordovician, approximately 483 mya (*Rota-Stabelli et al., 2013*). The transition from sea to land meant that molecules needed to be detected in gas phase instead of aquatic solution. Therefore, the olfactory system of a hexapod ancestor had to adapt to the terrestrial conditions and detection of volatile, air-borne chemicals. One proposed hypothesis has been that Orco and ORs of the insect type are an adaptation to this terrestrial lifestyle (*Robertson et al., 2003*; *Krång et al., 2012*). To reconstruct an evolutionary scenario for insect ORs, we investigated species belonging to different ancient insect orders, including Archaeognatha (jumping bristletails) and Zygentoma (silverfishes and firebrats), and a neopteran insect belonging to the Phasmatodea (leaf and stick insects) as so far not analyzed control group using morphological, electrophysiological and molecular techniques.

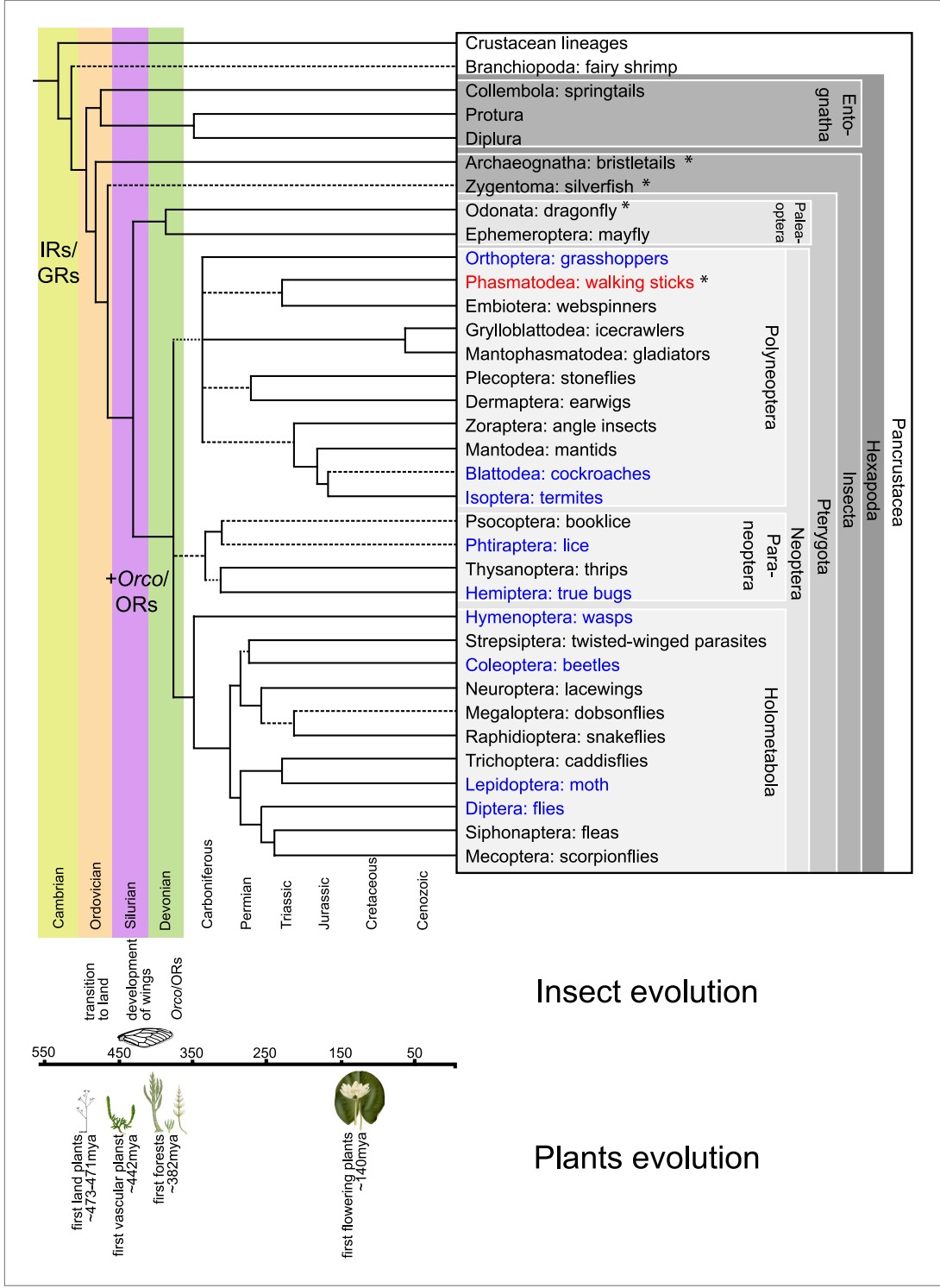

**Figure 1**. Hexapodan phylogeny. Phylogeny was adapted from *Trautwein et al. (2012)*. Timescale was adjusted for higher level taxa based on *Rota-Stabelli et al. (2013)*, for Holometabola according to *Wiegmann et al. (2009)* and the remaining groups based on their fossil record (http://insects.about.com/od/evolution/a/Timeline-of-Fossil-Insects-by-Order.htm), in order to correlate important events in plant and insect evolution with the emergence of insect olfactory receptors. IRs and GRs are known to be much older than insects (*Peñalva-Arana et al., 2009*; *Croset et al., 2010*), however, ORs and Orco have evolved during the evolution of insects and cannot be found outside the insect clade (*Peñalva-Arana et al., 2009*). Insects with a described OR/Orco-based olfactory system

*Figure 1. Continued on next page*

*Figure 1. Continued*

were highlighted in blue, whereas species were *Orco* was described in this study were colored in red. All orders investigated in this study are labeled by an asterisk. Our data suggests the evolution of the coreceptor Orco after the bristletails split from its last common ancestor with the remaining insects. However, an olfactory system that relies both on ORs and *Orco* seems to have evolved after the emergence of wings.

## Results

Our first step was to analyze the evolutionary ancestry of the insect olfactory system by assessing its complexity in each of three non-holometabolan insects.

To correlate OSN responses with type of sensillum (with pores and grooves) identified in SEM studies of the antennae, we investigated the morphological and physiological characteristics of olfactory sensilla and their olfactory sensory neurons.

### Morphology and physiology

On the antennae of *L. y-signata* the only putative olfactory sensilla were porous olfactory basiconic sensilla (*Figure 2B–E*). These sensilla were arranged in a pattern that is highly stereotypical between antennal modules composed of 5–12 annuli, with annuli typically containing zero-to-four *Sensilla basiconica* (*Missbach et al., 2011*). Responses to all tested chemical classes of odors, including acids, alcohols, aldehydes, esters, and ketones, were recorded from OSNs housed in these sensilla using the single sensillum recoding measurements (SSR) (*Figure 3*, uppermost heat map). Based on the response profile, spontaneous activity, and colocalization inside the same sensillum, we identified 12 OSN types, present in five functional basiconic sensillum types. Out of the 12 OSN types, only seven responded to odors tested; two exclusively to acids, while five responded with a similar activity rate to acids or amines and to other odors. OSNs belonging to this second class were broadly tuned and exhibited relatively low spiking activity. In general, OSN classes displayed a low baseline activity with about 1 to 7 spikes/s, with Lys-ab2A that had a spontaneous activity of more than 25 spikes/s as the only exception. Only rarely was an increase in spiking rate of more than 60 spikes per second recorded, even for the best identified ligands (*Figure 3— source data 1*). No responses were obtained for ammonia or pyridine. Coeloconic-like sensilla, s-shaped trichoid sensilla, and chaetic sensilla did not display any morphological features indicating olfactory function and did also not respond to any odor tested (*Missbach et al., 2011*; data not shown). In conclusion, 7 OSN types that were all housed in basiconic sensilla responded to a wide spectrum of odor molecules.

The morphology of the zygentoman antenna and its sensilla was similar to that of *L. y-signata*, with the presence of grooved sensilla as the only exception (*Figure 2G*; *Adel, 1984*; *Berg and Schmidt, 1997*). Five different functional types of olfactory sensilla were present (*Figure 3*: three porous, two grooved *s. basiconica*, the latter are indicated by blue caption). In contrast to *L. y-signata*, a nascent functional and spatial separation of the detection of amines and acids, and ketones and alcohols appeared in *T. domestica*. The former primarily elicited responses in OSNs of grooved sensilla, while less polar ones were mainly detected by porous sensilla. However, most of the OSNs in porous sensilla exhibited broad tuning and responded to at least one of the tested acids or amines as well.

We then turned to a neopteran insect. Unlike the other analyzed species, the leaf insect *P. siccifolium* displayed a strong sexual antennal dimorphism, with males having very long antennae covered with trichoid sensilla (*Figure 2L*), and the females very short antennae without trichoid sensilla (*Figure 2K*). In comparison to the wingless insects, the response repertoire of the leaf insect was much more diverse, with a total of 23 different functional sensillum types as identified by SSR recordings (*Figure 3*). No responses were obtained from trichoid sensilla, but since they were only present on the male antennae they could be involved in detection of an unknown volatile pheromone. In all cases, reported detection of volatile pheromones in insects is dependent on very specific ORs. Taken together these data suggest that leaf insects have a much broader response repertoire with a higher number of different OSN types than the more basal species we analyzed; apparently the number of olfactory receptors has increased. It also seems likely that at least the leaf insect makes use of ORs in odorant detection.

### An antennal and maxillary palp transcriptome

We generated expansive antennal transcriptome datasets of the three insect species, employing a bioinformatics-based approach to identify Orco, ORs, GR, and IRs. In a second transcriptome of

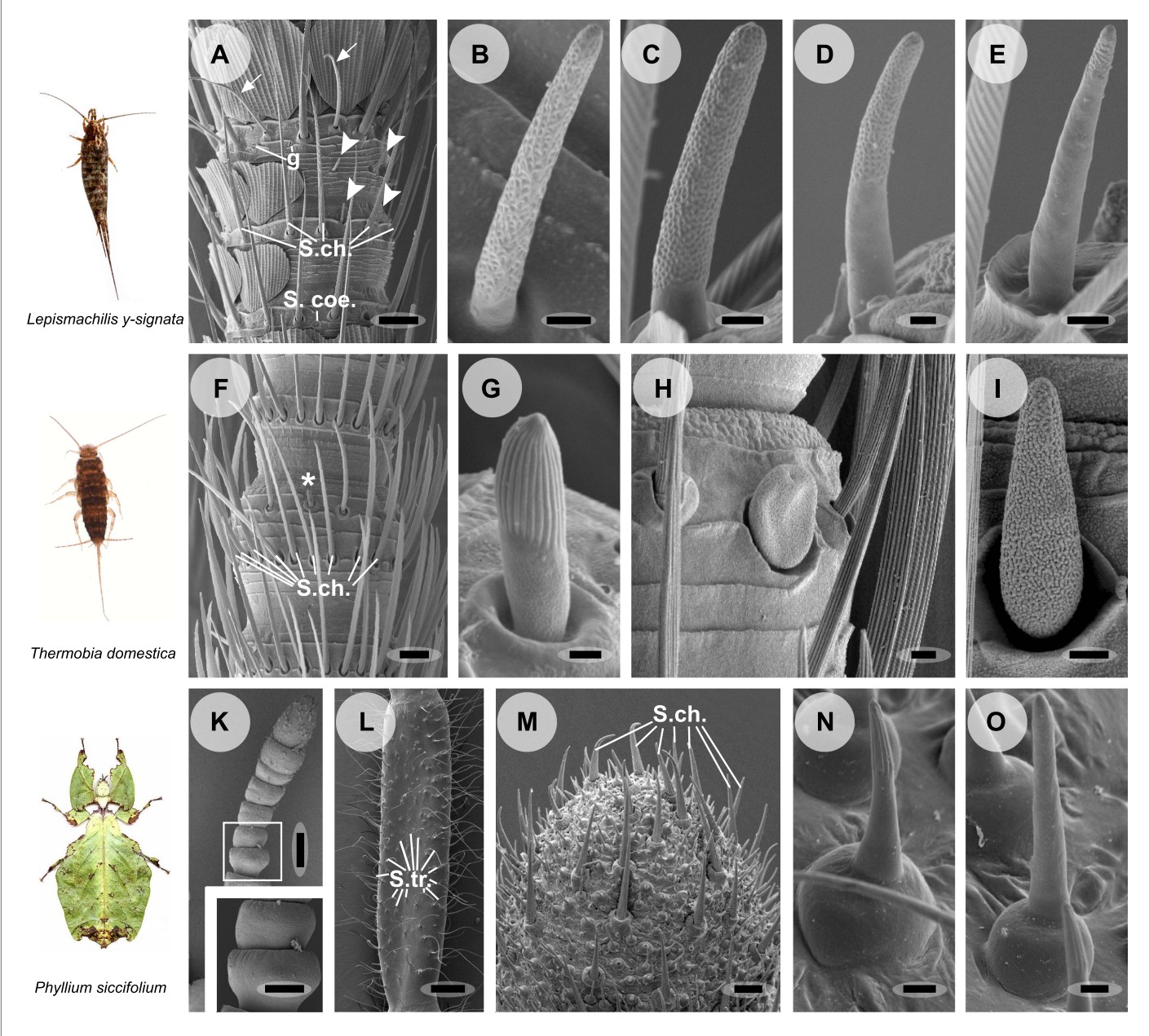

**Figure 2**. Olfactory sensilla on the antennae of *L. y-signata* (A–E), *T. domestica* (F–I) and *P. siccifolium* (K–O). Animals are depicted next to the corresponding antennal SEM images. (**A**) Detailed view of the antennae of *L. y-signata*. The proximal part of the antennae is not only covered with sensilla, but also scales. Glands (g) are highly abundant on the antennae. Many mechanosensory sensilla (S.ch.: Sensilla chaetica) were arranged in circles on the antennal segments. On some antennal segments gustatory sensilla (arrows) can be found between the S.ch (for further information read *Missbach et al., 2011*). Very rarely zero to four olfactory Sensilla basiconica were identified per segment, in a mostly redundant pattern on the antennae with similar numbers of olfactory sensilla and sensilla types on each antennal segment. Antennal segments are separated by antennal breaking points. The pattern of sensilla is modulated by increasing the number of annuli of a segment through molting. (**B–E**) Different morphological types of basiconic sensilla. No grooved sensilla/olfactory coeloconic sensilla were identified on the antennae. Only small pegs surrounded by a cuticular wall (s. coe.; referred as coeloconica-like sensillum, *Bockhorst 1988*) were located on the antennae. These sensilla are not olfactory (for detailed external morphology see *Missbach et al., 2011*). (**F**) Detailed view of the antennae of *T. domestica*. The antennal organization is similar to the bristletail, with antennal breaking points and lifelong molting. The most abundant sensilla on the antennae again are mechanosensory S.ch.; beside those gustatory and olfactory sensilla are distributed in a species-specific modular manner over the antennae. (**G**) In contrast to *L. y-signata*, grooved sensilla can be found on the antennae of *T. domestica*. (**H** and **I**) Different morphological types of basiconic sensilla. (**K** and **L**) Gender specific differences between a female (**K**) and a male (**L**) antennae of *P. siccifolium*. Female antennae are short and lack trichoid sensilla (S.tri.). They more or less lack sensilla on the proximal annuli, only the last

*Figure 2. Continued on next page*

*Figure 2. Continued*

two annuli are covered with a high number of olfactory and also some mechanosensory sensilla (S.ch.). (**M**) Male antennal tip. Similar to the distal female antennal annuli the highest density of sensilla can be found on the last annuli. (**N** and **O**) Both grooved and pored sensilla can be found on these segments. Scale bars: **A**: 50 µm; **B**, **C**, **D**, **E**, **H**, **I**, **N**, **O**: 2 µm; **F**: 100 µm; **G**: 1 µm; **K**, **L**: 200 µm; **M**: 20 µm.

*L. y-signata* also maxillary palp RNA was included. In total 99'504'815 reads were generated for the two *L. y-signata* chemosensory transcriptomes, out of which 77'060'687 were paired end reads. In addition to the transcriptomes of chemosensory tissues, we sequenced pooled RNA of whole bodies and heads resulting in 25'242'666 reads. This data set was analyzed separately. 27'704'231 and 30'762'777 reads were generated for antennae of *T. domestica* and *P. siccifolium*, respectively (detailed information about transcriptomes and assembly parameters can be obtained from the 'Material and methods' section and *Table 1*).

## No ORs or Orco were found in the transcriptome of *L. y-signata*

The transcriptome data sets were manually screened for genes encoding proteins putatively involved in insect olfaction, including ORs, Orco, GRs, and IRs (number of identified contigs are given in *Table 2*).

Neither OR- nor Orco-coding transcripts were identified in the transcriptomes of *L. y-signata* using BLAST and HMM domain profile searches as described in the 'Material and methods' section. Custom HMMR-profiles directed against conserved regions of Orco proteins also failed to identify any Orco-related sequences in the bristletail transcriptome. We discovered five GR candidates. MSA analysis of these together with ORs and GRs of various insect species and the *Daphnia* GRs always confirmed the position of the *L. y-signata* GR candidates within the GR and not the OR family (*Figure 4A*, *Figure 4—source data 1*, *Figure 4—source data 2*, *Figure 4—source data 3*, *Figure 4—source data 4*, *Figure 4—source data 5*). Since expression levels of gustatory receptors are very low even in gustatory tissue (*Clyne et al., 2000*; *Scott et al., 2001*), we argue that ORs or at least Orco should be represented in the large, sensory tissue-specific transcriptome data set of *L. y-signata* if they are indeed part of the olfactory system in the species.

## The three Orco-paralogues of *T. domestica*

In contrast to *L. y-signata,* three different Orco-related sequences were identified in the transcriptome of *T. domestica.* All candidates were cloned as full-length coding sequences using RACE-PCR. The three sequences displayed different similarities to the Orco sequence of *D. melanogaster*, one sequence shared 45.8%, one 35.1%, and the third 24.4% sequence similarity at the amino acid level. Orco was the protein most similar to all three Orco candidate sequences (*Figures 4B and 5*), although some of the key amino acids of the coreceptor are substituted at least in TdomOrco3 (*Wicher et al., 2008*; *Sargsyan et al., 2011*; *Nakagawa et al., 2012*; *Kumar et al., 2013*; highlighted in alignment *Figure 5*). Apart from the Orco variants, no OR-related sequences were identified, but 9 contigs for GR candidates were found that were assigned to seven GRs, including three candidates close to full length or full length and four additional fragments (*Table 2* and *Figure 4A*).

## Normal OR/*Orco* in the leaf insect

In the transcriptome data set of *P. siccifolium*, both various OR-related sequences and a single *Orco* sequence were detected (*Table 2*). The exact number of OR genes was hard to ascertain since some of the contigs were too short and did not show sufficient sequence overlap in a multiple sequence alignment (MSA) to be confidently identified as independent. However, in total, we identified 30 gene fragments coding ORs, indicating that the transcriptomic approach chosen was applicable to our question, successfully identifying both Orco and ORs in *P. siccifolium.*

### *Orco* expression in *T. domestica*

Considering that for all other insects analyzed so far one Orco is the norm, the appearance of three *Orco* candidates in *T. domestica* is highly unusual. We thus assessed the expression of the three candidates in different tissues using RT-PCR. For all three Orco types expression was limited to the antenna (*Figure 6*). To further assess the expression, we used in situ hybridization employing an antisense probe of one of the coreceptors. This led to staining of single cells below one or two basiconic sensilla of an antennal subsegment (*Figure 7*), suggesting that TdomOrco1 might indeed be expressed in OSNs. However, only one neuron per sensillum was stained. No signals were obtained when using a sense probe for TdomOrco1 (*Figure 7—figure supplement 1*).

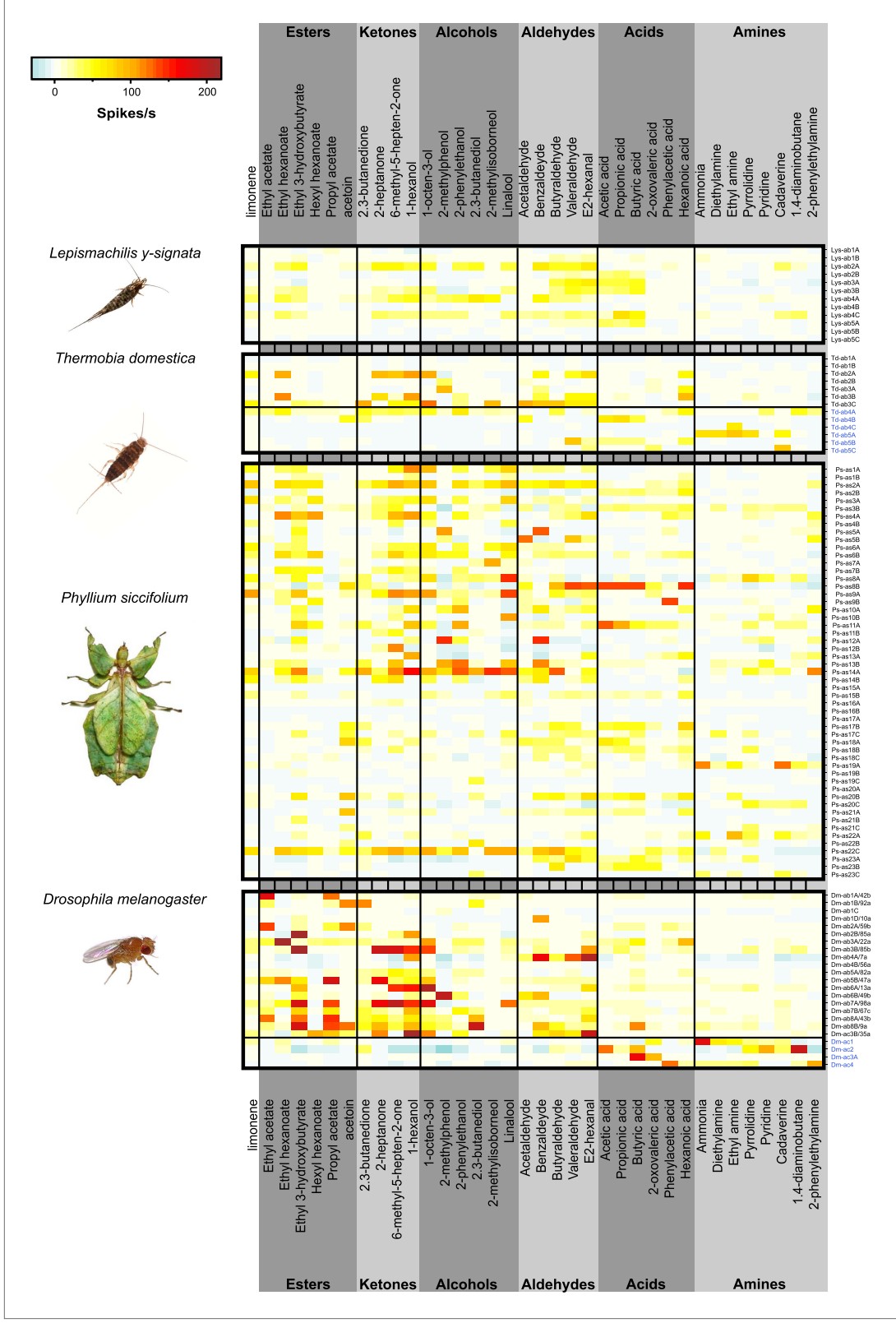

**Figure 3**. Color coded response profiles of *L. y-signata*, *T. domestica*, *P. siccifolium* and *D. melanogaster*. Spikes are sorted by neurons, with the exception of ac1, ac2, and ac4 of *D. melanogaster* where spike sorting was not possible. Means over 5 to 23 recordings were used as basis for visualization (source data are given in *Figure 3. Continued on next page*

*Figure 3. Continued*

*Figure 3—source data 1*). The same color code was used for all species, ranging from highest to lowest encountered change in activity. Neurons in grooved sensilla are indicated by blue letters (ac). For *L. y-signata* responses to odors were only obtained from neurons in porous sensilla (ab). A separation between porous and grooved sensilla was not possible for *P. siccifolium*. Sensilla were classified as antennal sensillum (as). *L. y-signata* neurons are mostly broadly tuned with comparable low change in spiking activity. For *P. siccifolium* a total of 23 different functional sensillum types were identified in SSR recordings (in comparison five in *L. y-signata*, five in *T. domestica*) suggesting that leaf insects have a broader response repertoire.

The following source data are available for figure 3:

**Source data 1**. Excel file of mean responses and baseline firing rate of the different OSN classes of *L.y-signata*, *T. domestica, P.siccifolium,* and *D. melanogaster*.

## Only IRs in *L-y-signata*

As none of the experiments gave a hint for the existence of any OR or Orco-related sequence in the bristletail transcriptome, we focused on the second olfactory receptor family of insects, the IRs. Although we could not identify any OR sequences in the transcriptome, a high number of putative glutamate receptor coding contigs was identified (*Table 2*). However, only five candidate iGluRs and 14 candidate IRs appeared to be real unigenes, possessing at least two of the three transmembrane domains. Some candidate sequences were extended in 3'-direction using RACE-PCR with antennal cDNA as template, allowing verification of unigene status and antennal expression. In MSA and phylogenetic analysis, the identified IRs grouped with DmelIRs (*Croset et al., 2010*). Among the identified putative LsigIRs were orthologues of the *D. melanogaster* coreceptors IR25a and IR8a, as well as one receptor similar to IR76b (*Figure 8A*, *Figure 8—source data 1*, *Figure 8—source data 2*, *Figure 8—source data 3*, *Figure 8—source data 4*, *Figure 8—source data 5*). As in other IRs (*Benton et al., 2009*) one or several key amino acids in the predicted glutamate binding domains were absent in the non-coreceptor IR candidates and LsigIR76b (*Figure 8B*). 7 out of 14 LsigIRs group close to a cluster of *D. pulex* IRs and the antennal IRs IR21a and IR68a of *D. melanogaster*, with no clear relationship to one or the other. None of the *Lepismachilis* IR candidates grouped with the 'divergent' *Drosophila* IRs.

We then performed fluorescent *in situ* hybridization with RNA probes directed against the IR coreceptor candidates (*Figure 9*). Antisense probes of IR25a and IR8a led to labeling of one to three OSNs underneath basiconic sensilla (*Figure 9—figure supplement 1*). In control experiments with sense probes, or without any probe, no staining was obtained (*Figure 9—figure supplement 2*). The pattern of expression of IR coreceptors in OSNs of *L. y-signata* indicates that most OSNs are covered by this gene family.

All experiments thus indicate that the olfactory system of this species employs other receptors like IRs or GRs, with no ORs or Orco present.

## Discussion

Insects provide us with an excellent opportunity to study groups of animals that have retained ancestral characteristics and understand how the specific building blocks in olfaction have evolved in both insects and other animals. Consequently, we selected insects at crucial positions of the phylogenetic tree with a functional olfactory system adapted to terrestrial conditions and detection of volatile chemicals. This species collection provides an excellent model to study the early evolution of the insect olfactory system.

To address which receptors are involved in odor detection in these insects and in basal insects in general, we applied several different approaches. Based on our transcriptome data sets, we suggest a stepwise evolution of the Orco/OR complex with Orco having evolved in the lineage of Dicondylia (Zygentoma + Pterygota) and the functional complex of Orco and ORs emerging within the pterygote insects (this study, *Clyne et al., 1999*; *Gao and Chess, 1999*; *Smadja et al., 2009*; *Vosshall et al., 1999*; *Robertson and Wanner, 2006*; *Kirkness et al., 2010*). Although it is impossible to completely rule out the presence of ORs, none of our extensive experiments led to the identification of either ORs or Orco in the bristletail *L. y-signata*. The well-established conservation of the *Orco*

**Table 1.** Technical overview of transcriptomes (study accession: PRJEB5093, study unique name: ena-STUDY-MPI CE-12-12-2013-15:03:23:860-31)

| Organism | Sequencing technique | Number of reads | Number of contigs above 400 bp | N50 | Average length of contigs | Tissue | Sample accession | Secundary accession | Sample unique name |
|---|---|---|---|---|---|---|---|---|---|
| *Lepismachilis y-signata* | HiSeq2000 (Illumina) | 22'444'128 | 68'984 | 1'179 | 1'000 | antennae and palps | ERS384175 | SAMEA2276780 | Lysig1 |
| | HiSeq2500 (Illumina) | 77'060'687 paired end | | | | antennae | ERS384176 | SAMEA2276781 | Lysig2 |
| | HiSeq2000 (Illumina) | 25'242'666 | 37'860 | | 857 | heads, whole bodies | ERS399748 | SAMEA2342071 | LysigMix1 |
| *Thermobia domestica* | HiSeq2500 (Illumina) | 27'704'231 paired end | 31'172 | 1'349 | 1'070 | antennae | ERS384177 | SAMEA2276782 | Tdom1 |
| *Phyllium siccifolium* | HiSeq2500 (Illumina) | 30'762'777 paired end | 34'653 | 1'890 | 1'305 | antennae | ERS384178 | SAMEA2276783 | Psic1 |

**Table 2.** Number of candidate contigs (not unigenes) for the different gene families identified in the transcriptomes of the different species

| Organism | Orco | ORs | GRs | IRs |
|---|---|---|---|---|
| *Lepismachilis y-signata* | – | – | 7 (5 above 400 bp) | 17 (16 above 400 bp) |
| *Thermobia domestica* | 6 (1 above 400 bp) | – | 9 (3 above 400 bp) | 19 (9 above 400 bp) |
| *Phyllium siccifolium* | 1 (1 above 400 bp) | 30 (16 above 400 bp) | 6 (2 above 400 bp) | 32 (19 above 400 bp) |

coding gene through evolution suggests that it is highly unlikely that we missed it. We did, however, identify a number of IRs, including the IR coreceptors IR25a, IR8a, and IR76b in the *L. y-signata* antennal transcriptome. FISH allowed us to visualize expression of the IR co-receptors in a large number of OSNs associated with basiconic sensilla. Based on these results we propose that the olfactory system of *L. y-signata* is not based on ORs.

In insects, different sensillum types house OSNs typically responding to different sets of odors. In *D. melanogaster* IRs are the functional receptor type of OSNs in double-walled coeloconic sensilla, and ORs are predominantly expressed in OSNs housed in single-walled basiconic and trichoid sensilla (*Hallem et al., 2004*; *Silbering et al., 2011*). It follows that this organization cannot exist with just one sensillum type present, as is the case in Archaeognatha (*Berg and Schmidt, 1997*; *Missbach et al., 2011*) and older hexapod taxa as the Collembola (*Altner and Prillinger, 1980*). The oldest insect taxon where double-walled sensilla were investigated is Zygentoma, which have both single-walled basiconic sensilla with pores and double-walled sensilla with spoke channels (*Berg and Schmidt, 1997*). Coeloconic sensilla differ dramatically from the single-walled trichoid and basiconic types in both wall structure and in internal environment. The coeloconic structure has been thought to be a prerequisite for IR function (*Benton et al., 2009*; *Guo et al., 2014*). However, in the Archaeognatha we find that IRs are most likely located in OSNs of Sensilla basiconica. IRs might thus have evolved in a single-walled sensillum and did not find their modern, coeloconic environment until neopteran insects evolved.

In the bristletail *L. y-signata,* we found that many of the OSNs are very broadly tuned, responding to volatiles with several different functional groups at higher doses. However, broadly tuned receptors might not have high affinities. By counting and integrating molecules over longer times, OSNs could include even low-probability binding events in generating their response (*Firestein, 2001*). This might also mean that the system does not have a high temporal resolution, which seems to be a fair trade-off for a walking insect that lives in its substrate.

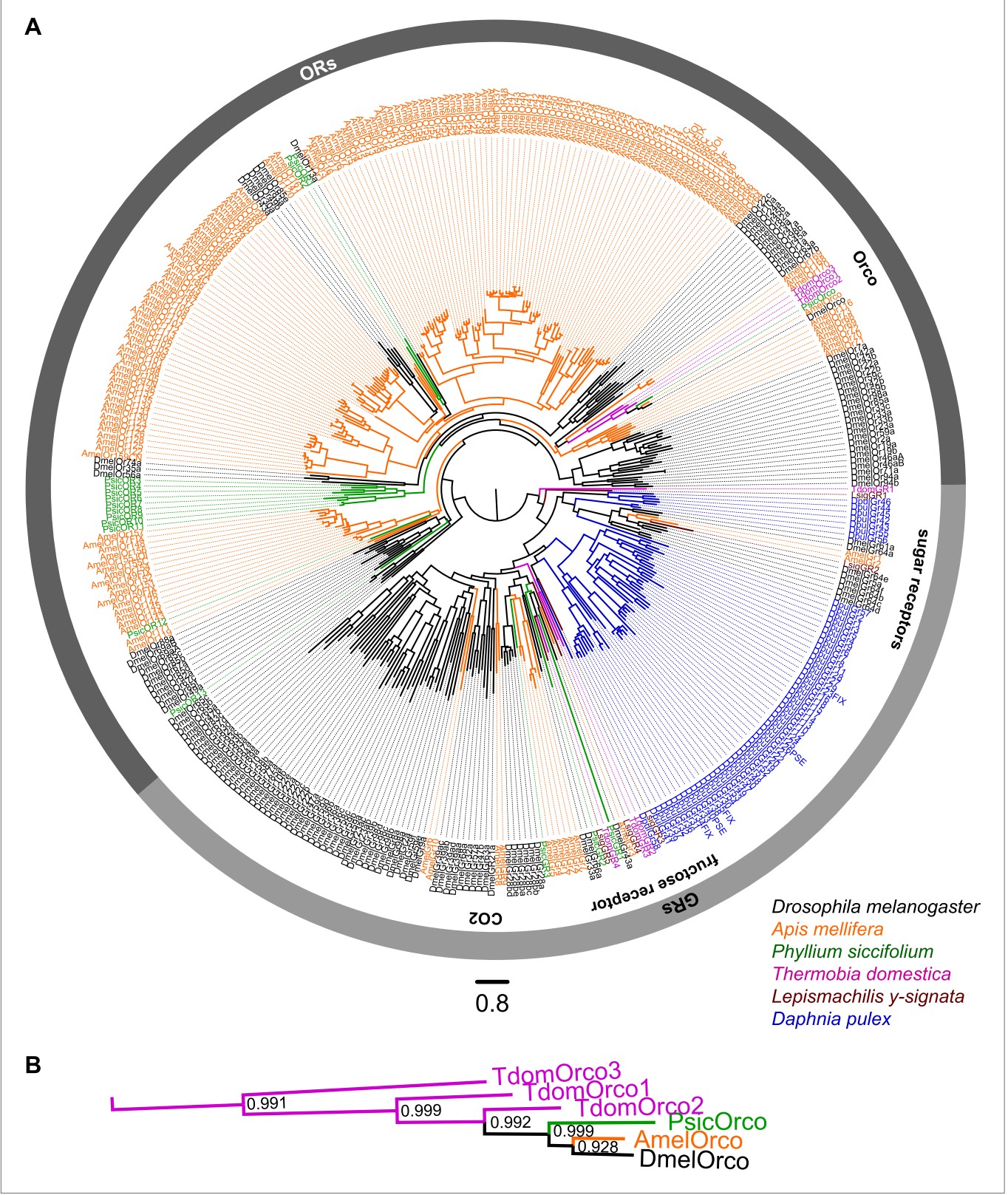

**Figure 4**. ORs and GRs of *L. y-signata, T. domestica,* and *P. siccifolium*. (**A**) Dendrogram displaying the relationship of identified OR and GR candidates of *L. y-signata, T. domestica,* and *P. siccifolium* to *D. melanogaster* (**Clyne et al., 1999**; **Gao and Chess, 1999**; **Vosshall et al., 1999**) and *Apis mellifera* (**Robertson and Wanner, 2006**) GR and OR proteins, and *Daphnia pulex* GRs (**Peñalva-Arana et al., 2009**). The dendrogram was determined by *Figure 4. Continued on next page*

*Figure 4. Continued*

maximum likelihood analysis of a MAFFT-Alignment using FastTree2. All *L. y- signata* candidates group within the GRs. Only candidates with a translated amino acid sequence longer than 120 amino acids and overlap in multiple sequence alignment were taken for analysis, since ORs and GRs are highly divergent and only unigenes should be included in the analysis (all candidate OR and GR sequences of *L. y-signata*, *T. domestica* and *P. siccifolium* are given in *Figure 4—source data 1* for amino acids and *Figure 4—source data 2* for nucleotide sequences). For *T. domestica*, we identified three different variant Orco types that were included in the analysis as full length translated amino acid sequences. (**B**) Blow-up of the dendrogram showing the support values for the coreceptor subgroup. The whole group is well supported.

The following source data are available for figure 4:

**Source data 1**. Amino acid sequences of putative olfactory and gustatory receptors of *L. y-signata*, *T. domestica,* and *P. siccifolium*.

**Source data 2**. Nucleotide sequences of putative olfactory and gustatory receptors of *L. y-signata*, *T. domestica,* and *P. siccifolium*.

**Source data 3**. MAFFT-alignment of OR and GR candidates of *L. y-signata*, *T. domestica*, *P. siccifolium* and *D. melanogaster* (***Clyne et al., 1999***, ***Gao and Chess, 1999***, ***Vosshall et al., 1999***) and *Apis mellifera* (***Robertson and Wanner, 2006***) GR and OR proteins, as well as *Daphnia pulex* GRs done.

**Source data 4**. FastTree file resulting from the MSA of *Figure 4—source data 3* (can be opened with FigTree).

**Source data 5**. Tree file resulting from the MSA of *Figure 4—source data 3* containing node support values (can be opened e.g., with Adobe Illustrator).

The response spectrum of *Drosophila* IRs is much narrower than the responses we find in the bristletail. If IRs are the only olfactory receptor type in basal insects they should exhibit a broader spectrum of possible ligands, including acids, aldehydes, alcohols, but also esters and ketones, as revealed in our physiological measurements. One additional observation in the bristletail is that many of those neurons have a broad overlap in their response spectra. One hypothesis to explain an IR-based olfactory system in *L. y-signata* would be very broad tuning of single receptors, another that the selectivity of OSNs could be regulated by combinations of different IRs.

In *D. melanogaster*, one conserved IR (IR64a) is expressed in different subpopulations of sensilla in the third chamber of the sacculus (***Silbering et al., 2011***). Corresponding OSNs are activated either by free protons or organic acids and many other odors, including esters, alcohols, and ketones (***Ai et al., 2010***). Expression of this IR together with IR8a is both necessary and sufficient for sensitivity towards organic acids and other odors, but probably requires a different, until now unknown cofactor to mediate the specific response of OSNs to inorganic acids and $CO_2$ (***Ai et al., 2010***).

Alternatively, GR candidates could account for part of the non-neopteran olfactory setup, especially since it has been shown that GRs can add to the olfactory repertoire (***Tauxe et al., 2013***). Putative contact chemosensory sensilla are highly abundant on the antennae of *L. y-signata* (***Missbach et al., 2011***) and *T. domestica* (***Adel, 1984***). Both detection of sugars/amino acids (shown for *T. domestica*: ***Hansen-Delkeskamp, 2001***) and a proposed contact-pheromone (***Fröhlich and Lu, 2013***) likely involve GRs, indicating that involvement of the limited set of GRs beyond this scope is unlikely.

However, these data do not explain the presence of three different Orco variants in the firebrat. So far only one Orco orthologue has been identified in each studied insect species (e.g., ***Krieger et al., 2003***; ***Pitts et al., 2004***; ***Smadja et al., 2009***; ***Yang et al., 2012***). All *T. domestica* variants were found to be expressed in antennae, suggesting their involvement in chemosensation. TdomOrco3 even has an amino acid exchange of a functional important residue from asparagine to glutamic acid at position 466. This residue was demonstrated as critical for the ion channel function in *D. melanogaster*, where substitution of D466 with amino acids other than glutamic acid resulted in a substantial reduction in channel activity, but substitution to glutamic acid leads to an increase in sensitivity of the heteromeric receptor complex (***Kumar et al., 2013***). Additionally, this residue is highly conserved across insects (***Kumar et al., 2013***) including two of the three *T. domestica* Orcos (this study).

While the antennal expression argues for a potential involvement in chemosensation, the existence of three Orco types remains mysterious. It will be part of future studies to investigate if the Orco candidates form heterodimers with other receptors like GRs or with each other to build functional receptors or if they fulfill a channel function in other processes than olfaction.

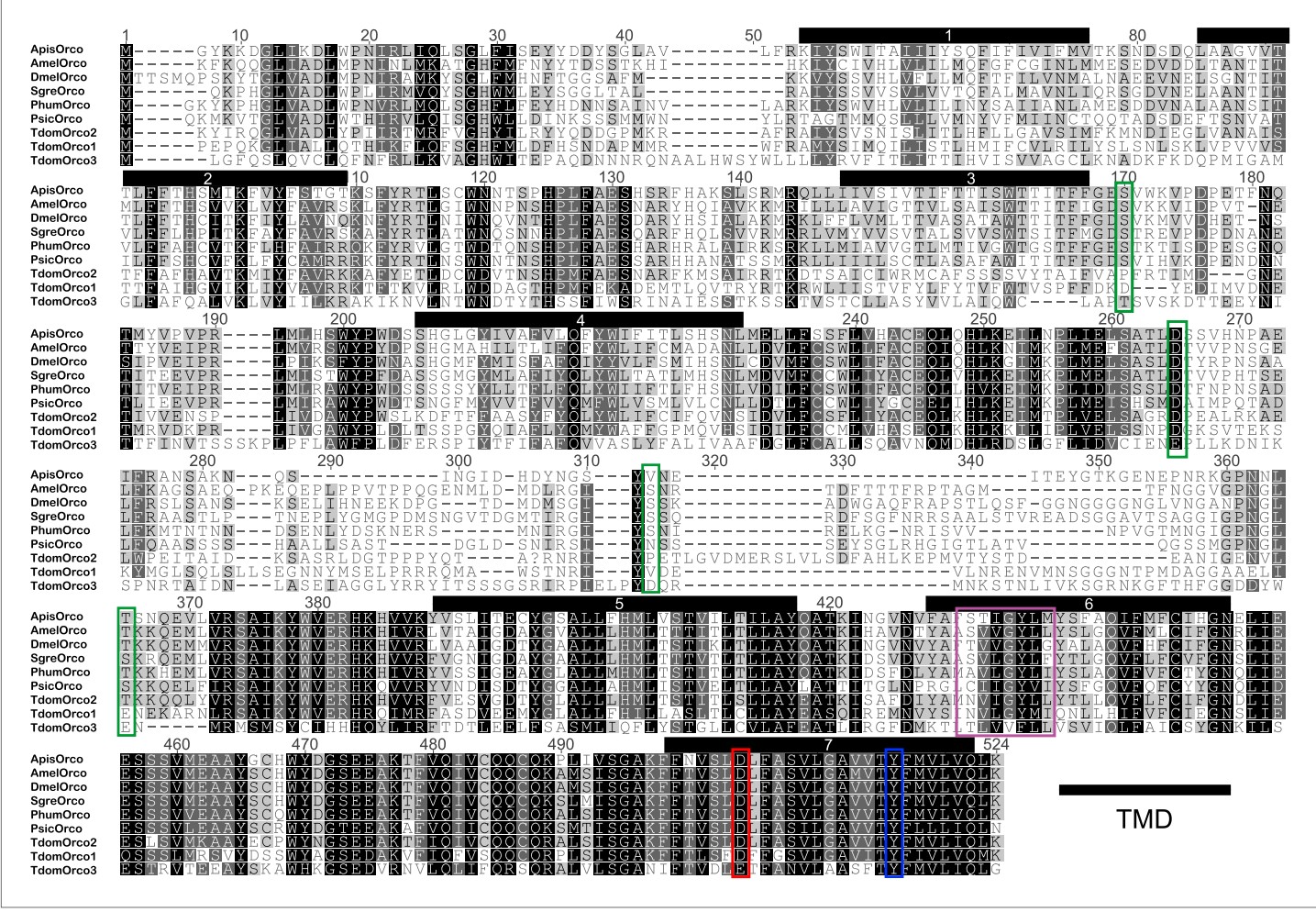

**Figure 5**. Multiple sequence alignment of *T. domestica* Orcos. Alignment of *T. domestica Orco*s with Orcos of *Acyrthosiphon pisum* (GI:328723530), *A. mellifera* (GI:201023349), *D. melanogaster* (GI:24644231), *Schistocerca gregaria* (GI:371444780), *Pediculus humanus corporis* (GI:242009783), *P. siccifolium* (this study). Important amino acids are highlighted in colored boxes (purple: effect on ion permeability, *Wicher et al., 2008*; green: phosophorylation sites for PKC of DmelOrco, *Sargsyan et al., 2010*; blue: affect spontaneous and evoked action potentials in receptor complex, *Nakagawa et al., 2012*; red: important residue for channel activity, *Kumar et al., 2013*).

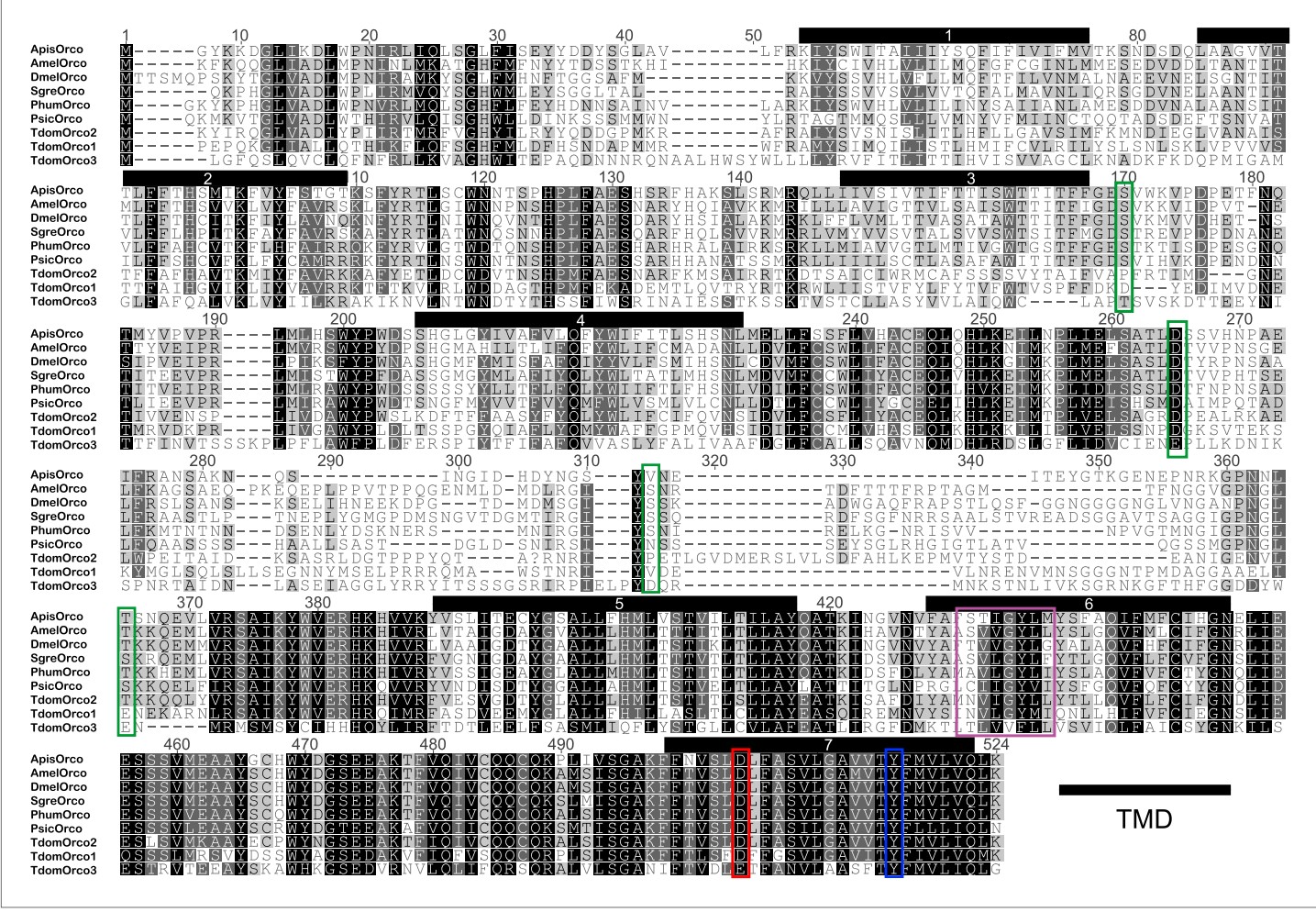

**Figure 6**. Expression of *T. domestica* Orcos. Using RT-PCR Orco expression was detected in the antennae (A) of *T. domestica*, but not in legs (L), heads without antennae and palps (H), and bodies (B). Primer sequences are given in *Figure 6—source data 1*.
The following source data are available for figure 6:

**Source data 1**. Primers and their properties used in this study.

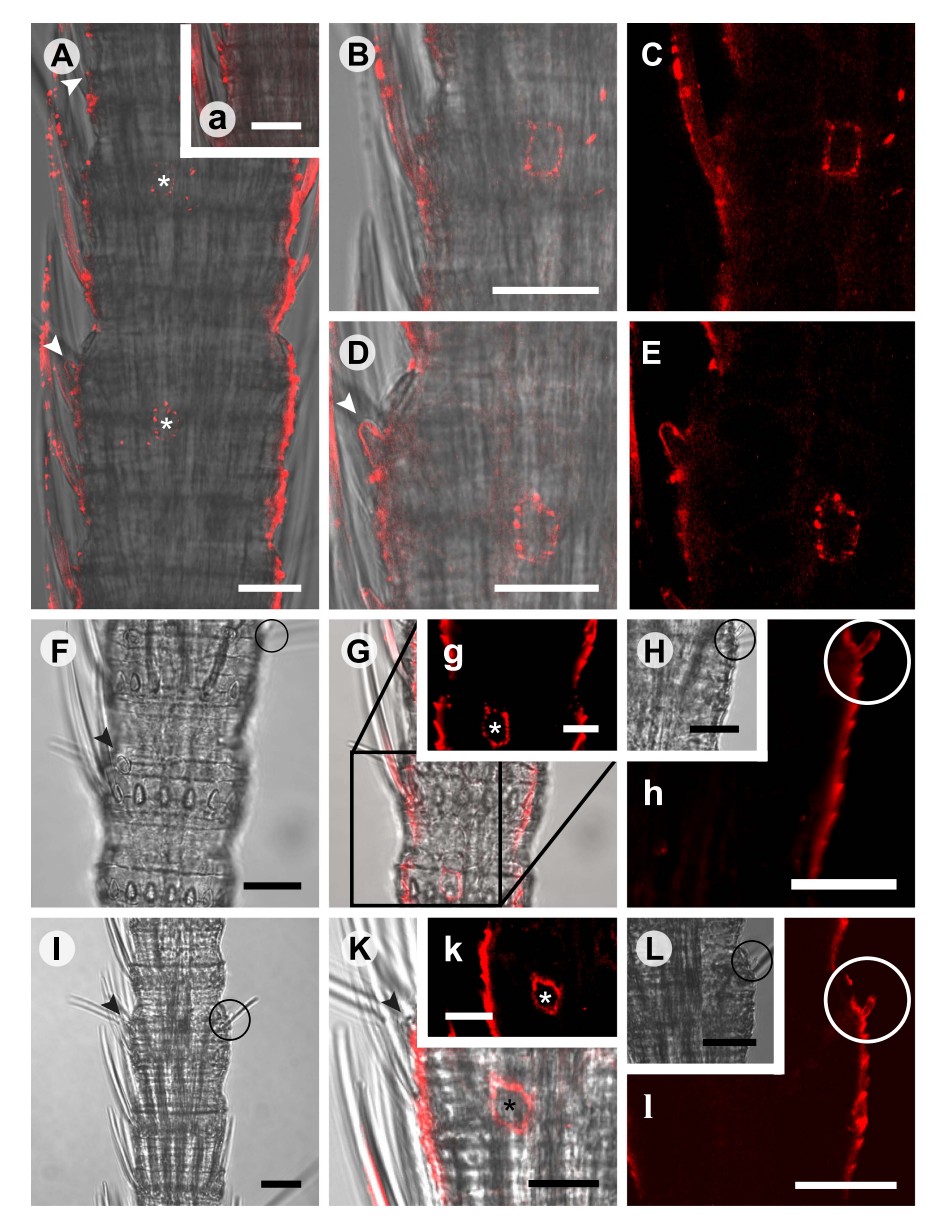

**Figure 7**. *In situ* hybridization on whole mount antennae of *T. domestica* using a Dig-labeled TdomOrco1 antisense probe. (**A**) Part of a *T. domestica* antenna. Combined image of fluorescent and transmitted light channel taken with cLSM. The positions of pored sensilla are indicated by arrowheads, with the upper sensillum displayed in the small box in the upper right corner. Labeled cell bodies are assigned by asterisks. (**B**–**E**) Single confocal planes through the antenna. Only a single soma close to each pored sensillum is labeled suggesting that only one neuron per sensillum expresses this Orco variant. In **B** and **D** some precipitate is visible. (**C** and **E**) Same image section as **B** and **D**, but without transmitted light. (**F**) Transmitted light image of a part of a second antenna. Location of a pored sensillum is again assigned by an arrowhead. A grooved sensillum indicated by a black circle is situated on the opposite side of the antenna. (**G**) Same part of the antenna taken with transmitted light and fluorescent channel. Again only one soma is labeled close to a pored sensillum. g: Only the Dig signal. Cuticle shows a strong autofluorescence on both sides. **H**, h: No signal was obtained close to a grooved sensillum. (**I**) Part of another antenna with a pored and a grooved sensillum on the same annulus. **K**, k: Image section from the part of the antenna close to the pored sensillum. A single soma is labeled by the probe. k: Only the fluorescent signal. **L**, l: No soma was labeled close to the grooved sensillum. For sense controls view ***Figure 7—figure supplement 1***. Scale bars **A**–**F**, **H**, **I**, **L**: 20 µm; g, **K**, k: 10 µm.

*Figure 7. Continued on next page*

*Figure 7. Continued*

The following figure supplements are available for figure 7:

**Figure supplement 1**. *In situ* hybridization on the antenna of *T. domestica* using sense probes directed against the TdomOrco1.

Altogether our data suggests that ORs evolved in insects after the emergence of Archaeognatha and Zygentoma, and therefore long after insects transitioned to a terrestrial lifestyle. At the time when flying insects occurred, the vegetation on earth was rapidly spreading and diversifying. ORs might not only increase the diversity of detected chemicals, but also allow the olfactory system to rapidly assess airborne odors. This is especially important for insects for which stimulus contact is very short and a fast response time is critical (*Getahun et al., 2012*). The oldest flying insect orders Odonta (dragonflies and damselflies) and Ephemeroptera (mayflies) were traditionally considered to be anosmic, lacking both a glomerular antennal lobe and mushroom body calyces (*Strausfeld et al., 1998*; *Farris, 2005*). Recent studies have shown that at least dragonflies have an aerial sense of smell (*Rebora et al., 2012*). However the small antennae and the low number of olfactory sensilla will make it even more challenging to identify putative ORs and Orco in antennal transcriptomes. ORs were definitely present in the last common ancestor of 'hemi'- and holometabolan insects at least 318–300 million years ago, with *Orco* present in both groups (this study, *Krieger et al., 2003*; *Pitts et al., 2004*; *Smadja et al., 2009*; *Yang et al. 2012*). The increasing dispersion of vascular plants together with the development of wings and a secondary wing articulation opened new and wider ranges of habitats and ecological niches for insects and the receptors to find them.

## Material and methods

### Animals

Different stages and sexes of *Lepismachilis y-signata* were collected at several locations around Jena (Germany). Animals were kept under normal light conditions and room temperature, in plastic boxes with paper towel on the ground, covered with bark with lichens, dried grassroots, and dead leaves of maple (*Acer campestre*, Sapindaceae). The boxes were moistened twice a week.

Firebrats of the species *Thermobia domestica* were obtained from a colony of the Botanical garden of Friedrich-Schiller University of Jena. Animals were maintained in a plastic container with paper towel on the bottom and egg cartons filled with cotton at around 25°C and 50–75% humidity, and were fed fish food (Zierfischflocke, TFH-Haimerl, Roding, Germany).

Different stages and sexes of *Phyllium siccifolium* were provided by the Institute of Systematic Zoology and Evolutionary Biology of the Friedrich-Schiller University of Jena. Animals were kept in a big gaze cage at 25°C and normal light cycle feeding on blackberry leaves. The substrate was moistened every second day.

### Physiology

#### Odorants

Pure odorants were diluted ($10^{-2}$) in hexane or in water as appropriate. Diluted odors (10 µl) were pipetted onto a small piece of filter paper (~1 cm$^2$) and placed inside a glass Pasteur pipette. For odorant application, a stimulus controller was used (Stimulus Controller CS-55, Syntech, Hilversum, The Netherlands).

#### Single sensillum recordings (SSR)

Adult animals were immobilized and the antennae were placed in a stable position. Sensilla were localized at 1000x magnification and the extracellular analog signals originating from the OSNs were detected by inserting a tungsten wire electrode in the base of a sensillum. The reference electrode was inserted into the eye or the body. Signals were amplified (10x; Syntech Universal AC/DC Probe), sampled (10,667.0. samples/s), and filtered (100–3000 Hz with 50/60 Hz suppression) via USB-IDAC connection to a computer (Syntech). Action potentials were extracted as digital spikes from the analog signal according to top–top amplitudes using Syntech Auto Spike 32 software. Neuron activities were recorded for 10 s, starting 2 s before a stimulation period of 0.5 s. Responses of individual neurons were calculated as the increase (or decrease) in the action potential frequency (spikes/s) relative to the

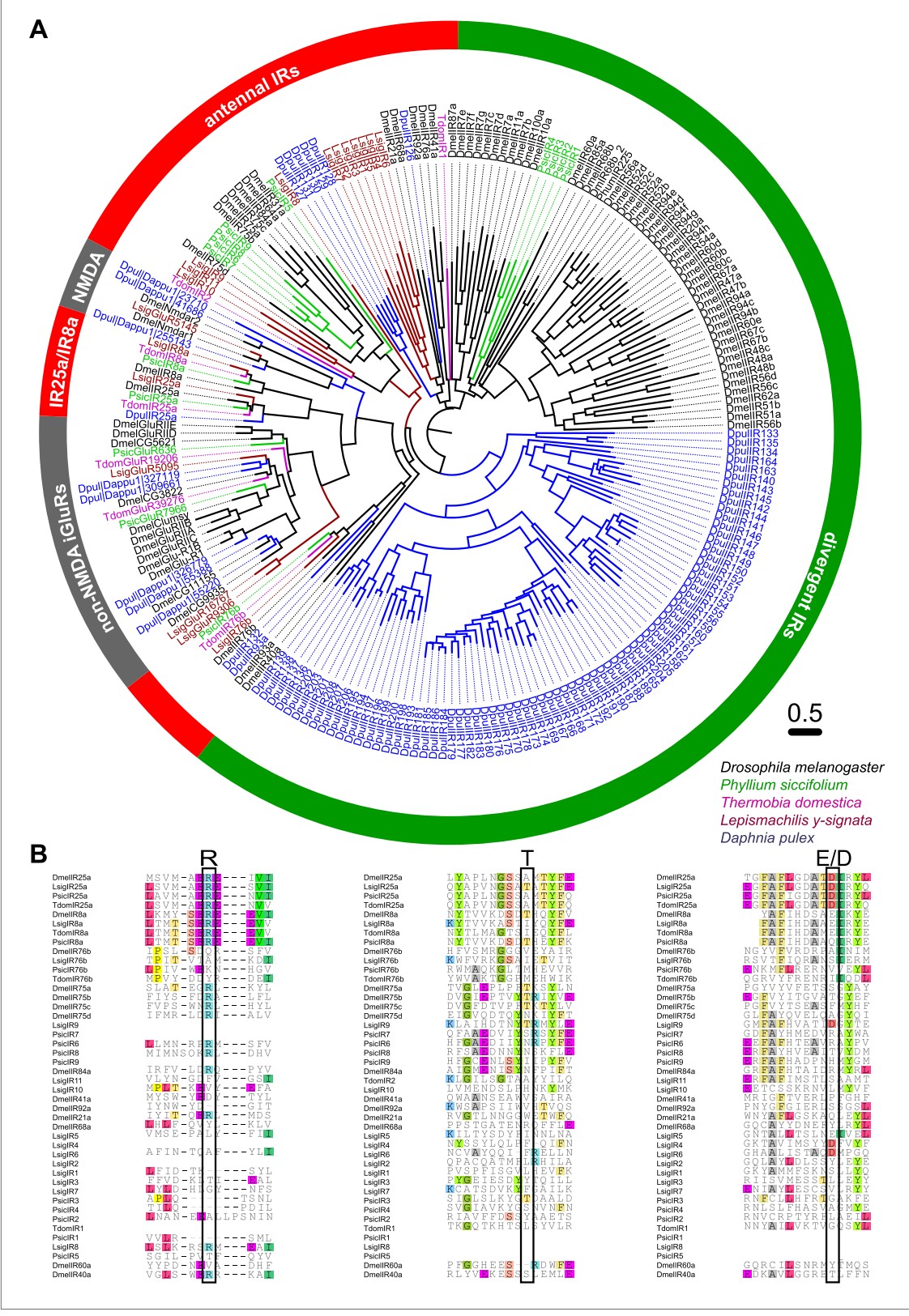

**Figure 8**. Ionotropic glutamate receptors of *L. y-signata*, *T. domestica*, and *P. siccifolium*. (**A**) Analysis of the relationship between *L. y-signata*, *T. domestica*, *P. siccifolium*, *D. melanogaster* and *D. pulex* iGluRs and IRs (*D. melanogaster* and *D. pulex* sequences were sequences taken from **Croset et al., 2010**). Amino acid sequences were aligned using the MAFFT alignment tool plug-in in Geneious Pro 5.0.4 (BLOSUM72, gap open

*Figure 8. Continued on next page*

*Figure 8. Continued*

penalty: 1.53, offset value: 0.123, E-INS-i settings). The dendrogram was generated using maximum likelihood analysis with FastTree2. (All candidate IR sequences of *L. y-signata*, *T. domestica*, and *P. siccifolium* are given in *Figure 8—source data 1* for amino acids and *Figure 8—source data 2* for nucleotide sequences) (**B**) Excerpts of the alignment showing the predicted glutamate binding domains and key amino acids. Mutations in one or several of the key amino acids are a structural feature to distinguish between iGluRs and IRs, although they can be present in the coreceptors.

The following source data are available for figure 8:

**Source data 1**. Amino acid sequences of putative variant ionotropic glutamate receptors of *L. y-signata*, *T. domestica,* and *P. siccifolium*.

**Source data 2**. Nucleotide sequences of putative variant ionotropic glutamate receptors of *L. y-signata*, *T. domestica,* and *P. siccifolium*.

**Source data 3**. MAFFT amino acid alignment of iGluR and IR candidates of *L. y-signata*, *T. domestica*, *P. siccifolium, D. melanogaster,* and *D. pulex* (*D. melanogaster* and *D. pulex* sequences were sequences taken from *Croset et al., 2010*).

**Source data 4**. FastTree file resulting from the MSA of *Figure 4—source data 3* (can be opened with FigTree).

**Source data 5**. Tree file resulting from the MSA of *Figure 8—source data 3* containing node support values.

pre-stimulus frequency. Sensilla were classified as basiconic, coeloconic, or trichoid based on morphological criteria. Further subdivision of distinct sensillum types was based on response profiles of all the OSNs housed within, independently from their possible olfactory receptor.

## SEM

Male and female antennae were cut at the base and fixed in glutaraldehyde. Antennae were dehydrated in an ascending ethanol series (70%, 80%, 90%, 96%, 3 × 100% ethanol, 10 min each), critical point dried (BAL-TEC CPD 030, Bal-Tec Union Ltd., Liechtenstein), mounted on aluminum stubs with adhesive film and sputter coated with gold on a BAL-TEC SCD005 (Bal-Tec, Balzers, Liechtenstein). Micrographs were taken with a LEO 1450 VP scanning electron microscope (Zeiss, Wetzlar, Germany).

## Molecular Biology and bioinformatics

### RNA extraction

Antennae and maxillary palps were cut off close to the base and were transferred to Eppendorf cups chilled over liquid nitrogen. RNA of different tissues, respectively antennae, palps, heads, whole bodies and juveniles (unscaled juvenile stadia) was isolated using TRIzol isolation following the manufacturer's instructions, but replacing chloroform with 1-bromo-3-chloro-propane. Total RNA was dissolved in RNase free water and total RNA quality and quantity measured using an Agilent Bioanalyzer (Agilent Technologies, Santa Clara, USA).

### Transcriptome sequencing

RNASeq was performed for *L. y-signata* RNA using the HiSeq 2000 (TruSeq SBS v5) Sequencing System from Illumina, utilizing the single read 100 bp (+7 index) technology at Eurofins MWG/Operon (Berlin). The resulting 22'444'128 reads were filtered for vector and linker sequences, as well as contaminants by Eurofins. A second RNASeq run for deeper sequencing was done using the HiSeq2500 at the Max Planck Genome centre in Cologne, resulting in 77'060'687 paired end reads of 100bp. Additionally to the transcriptomes of *L. y-signata* chemosensory tissues, a pooled transcriptome of whole body and head RNA was generated at Eurofins MWG/Operon (Berlin) using single read 100 bp (+7 index) technology.

Both *T. domestica* and *P. siccifolium* RNA was sequenced using the HighSeq2500 Sequencing system generating 27'704'231 paired end reads for *T. domestica* and 30'762'777 paired end reads of *P. siccifolium*. Before sequencing rRNA depletion was performed at the Max Planck Genome centre. Since the depletion did not work out for *L. y-signata*, a much deeper sequencing was performed in the second sequencing run as described above.

### Bioinformatics

Removal of duplicate reads and de novo assembly was performed with CLC Genomics Workbench 5.5 (CLCbio, Copenhagen, Denmark). Sequence databases were generated in Geneious Pro 5.0.4 (Biomatters, Auckland, New Zealand). Within these databases, we manually tBLASTn searched for

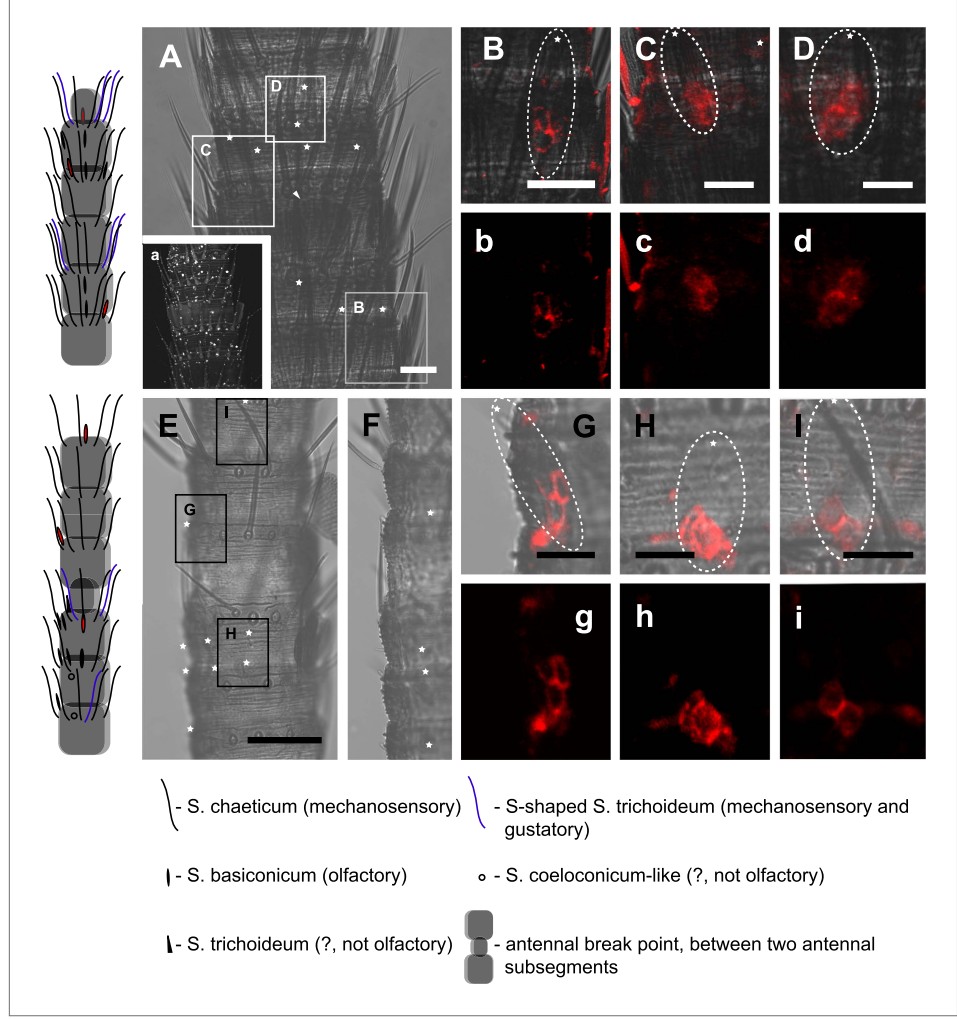

- S. chaeticum (mechanosensory)

- S-shaped S. trichoideum (mechanosensory and gustatory)

- S. basiconicum (olfactory)

- S. coeloconicum-like (?, not olfactory)

- S. trichoideum (?, not olfactory)

- antennal break point, between two antennal subsegments

**Figure 9**. *In situ* hybridization on the antenna of *L. y-signata* using a Dig-labeled LsigIR25a antisense probe. On the left: schematic drawings of the position of the different sensillum types on the particular antennal subsegment. The legend for the sensillum types is given below the confocal images. (**A**–**D**) Labeling of somata in a subsegment of an antenna. Mostly two somata were labeled with the probe. The positions of the somata were in line with the positions of basiconic sensilla, but not gustatory and mechanosensory sensilla. Ultrastructural investigation of basiconic sensilla of *Machilis* sp. (Archaeognatha) and *Lepisma saccharina* (Zygentoma) suggests that the sensory neurons are located in a distance of at least 25 μm from the sensillum base in the extension of the sensillum (***Berg and Schmidt, 1997***). Therefore, we concluded that the labeled somata correspond to neurons housed in basiconic sensilla. These sensilla were colored red in the drawing on the left. (**A**) Transmitted light overview with asterisks labeling basiconic sensilla. Image sections given in **B**–**D** are indicated by white boxes and the corresponding letters. a: Projection of confocal planes recorded with Argon laser at a wavelength of 488 nm to identify the position of basiconic sensilla. (**B**–**D**) Overlaid transmitted light and fluorescent images of labeled somata. b–d: Images without transmitted light channel. (**E**–**I**) Labeling of somata in a second antenna. Parts of two antennal segments that are separated by an antennal break point. The break point can be recognized by a thinner segment on the distal part of the antennae or by a special trichoid sensillum that is only present on the segment proximal to a breaking point. (**E** and **F**) Transmitted light images of the antenna. **E** is more from the top. Image sections given in **G**–**I** are indicated by white boxes and the corresponding letters. **F** is more central plane. Asterisks denote the location of a basiconic sensillum. (**G**–**I**) Overlaid confocal images of labeled neurons. Images are projections of three confocal planes. On some positions the cuticle is given a background signal. g–i: Images without transmitted light channel. Scale bars: **A**–**C**, **G**–**I**: 20 μm, **E**: 50 μm, **D**: 10 μm.

*Figure 9. Continued on next page*

*Figure 9. Continued*

The following figure supplements are available for figure 9:

**Figure supplement 1**. *In situ* hybridization on the antenna of *L. y-signata* using an antisense probe directed against the IR coreceptor IR8a.

**Figure supplement 2**. *In situ* hybridization on the antenna of *L. y-signata* using sense probes directed against the IR coreceptors IR25a, IR8a.

olfactory receptors (ORs), antennal ionotropic receptors (IRs), and gustatory receptors (CSPs). Templates for manual searches were the published amino acid sequences of the respective gene families of *Drosophila melanogaster*, *Bombyx mori*, *Pediculus humanus*, *Apis mellifera*, *Acyrthosiphon pisum*, and *Daphnia pulex*, as well as identified sequences of *L. y-signata*, *T. domestica*, and *P. siccifolium*.

Contigs with similarity to a member of these gene families were edited and subject to personal scrutiny of blast results, as well as further analysis. ORFs were identified and translated into amino acid sequence in Geneious Pro 5.0.4. Alignments with other members of the respective gene families were carried out using MAFFT (E-INS-I parameter set; *Katoh et al., 2005*). Dendrograms were calculated using maximum likelihood analysis with FastTree2 (*Price et al., 2009*; *Liu et al., 2011*) and displayed and edited with FigTree (http://tree.bio.ed.ac.uk/software/figtree). Candidates were named with the abbreviation for the gene family and ascending numbers with the exception of coreceptors, where a clear homology could be assigned. The body transcriptome of *L. y-signata* was independently screened for both ORs and Orco-related sequences.

Gene Ontology (GO) annotation was performed with Blast2GO (http://www.blast2go.com/b2ghome, *Conesa et al., 2005*).

## HMMR-design

HMMER v3.0 (*Eddy, 2011*) was used to construct HMM profiles based on a multiple sequence alignment of Orco sequences of *D. melanogaster*, *Apis mellifera*, *Tribolium castaneum*, and *Manduca sexta* resulting in three local HMM (83bDom_1: VKHQGLVADLMPNIRLMQMVGHFMFNYYS,

83bDom_4: TVEIPRLMIKSWYPWDAMHGM,

83bDom_5: DVMFCSWLLFACEQLQHLKAIMKPLMELSASLDTYRPNS) profiles and a global HMM profile. Profiles were used to search online against nr (http://hmmer.janelia.org/search/phmmer) to test the quality of the generated HMM profiles. Profiles were used subsequently to screen the antennal and maxillary palp transcriptome database of *L. y-signata* using the command line version of HMMER.

## cDNA synthesis for RT-PCR

SuperScript III First-Strand Synthesis System (Invitrogen, Life Technology, Grand Island, USA) was used for cDNA synthesis according to the manufacturer's instructions, including a DNAse digestion step.

## Receptor cloning

To validate and extend candidate sequences total RNA was purified using the Poly(A)Purist MAG Kit (Ambion, Life Technologies, Grand Island, USA). Synthesis of cDNA was performed using the SMARTer RACE cDNA Amplification Kit (Clontech, Mountain View, USA). Gene-specific primers were designed against receptor candidates (Primer3 v.0.4.0, Whitehead Institute for Biomedical Research and Oligo Calc version 3.26). RACE-PCR amplification was done according to the manufacturer's instructions.

## FISH

Biotin- and digoxigenin (DIG)-labeled sense and antisense probes targeting candidates were prepared using a T7/Sp6-Polymerase (ROCHE, Berlin, Germany) as per manufacturer's instructions, a Biotin RNA Labeling Mix 10x conc. (ROCHE) or DIG RNA Labeling Mix 10x conc. (ROCHE), and incubating 3 hr at 37°C. RNA was precipitated and washed once with 70% ethanol, dissolved in water and finally diluted in hybridization buffer. Probes were fragmented to a length of about 600 nucleotides (*Angerer and Angerer, 1992*).

Antennae of adult *L. y-signata* and *T. domestica* were cut off, shortly dipped in distilled water with Triton X-100 (Sigma Aldrich, St. Louis, USA) and fixed for 24 hr in 4% PFA (ROTH, Karlsruhe, Germany) in 1 M NaHCO$_3$ (Sigma Aldrich, pH 9.5). The antennae were washed in 1xPBS containing

0,03% TritonX100 and incubated in 0.2 M HCl (0.03% TritonX100) for 10 min. Afterwards, antennae were rinsed twice in 1xPBS (1% TritonX100) and autoclaved distilled water. After incubation in 2xSSC (3 M NaCl, ROTH; 0.3 M $C_6H_5Na_3O_7*2H_2O$, Sigma; pH 7.1) at 70°C a treatment with Proteinase K (1U/ml Proteinase Buffer) at 37°C for 30 min followed. The antennae were thoroughly washed in PBS and fixed again for 20 min. Fixative was washed away with PBS and antennae pre-hybridized in Hybridization Buffer for 8 hr at 55°C. Hybridization was performed at 55°C for 2 to 3 days. DIG-labeled probes were detected using an anti-DIG-conjugated antibody in combination with HNPP/FastRed (HNPP Fluorescent Detection Set, Roche), biotin-labeled probe using a TSATM Flouresin System. Preparations were analyzed using a Zeiss LSM510 Meta (Zeiss, Jena, Germany).

Due to the modular organization of the antenna, with compartments of a size varying between 5 and 12 annuli, and to the repetitive pattern of olfactory sensilla between the compartments, we did not need to map labeling of neurons along the whole antenna.

## Image processing

Contrast and false color images were optimized in Zeiss LSM Image Browser (Version 4,0,0,157). Further image processing, including cutting and image mode conversion was done in Adobe Photoshop CS4, figures were prepared in Adobe Illustrator CS4.

## Acknowledgements

We thank Christin Grossmann and Sascha Bucks for technical assistance in sample preparation and receptor cloning. The authors thank Renate Kaiser and Sandor Nietzsche (both Electron Microscopy Center Jena) for help with the scanning electron microscope, Richard Reinhardt and Liza Czaja (both Max Planck Genome Center) for support in transcriptome sequencing. We also thank Shannon Olsson and Dieter Wicher for critical comments on the manuscript.

## Additional information

### Funding

| Funder | Author |
| --- | --- |
| Max Planck Society | Bill S Hansson |

The funder had no role in study design, data collection and interpretation, or the decision to submit the work for publication.

### Author contributions

CM, Conception and design, Acquisition of data, Analysis and interpretation of data, Drafting or revising the article; HKMD, Conception and design, Acquisition of data, Analysis and interpretation of data; HV, MCS, Analysis and interpretation of data, Drafting or revising the article; AV, Drafting or revising the article, Contributed unpublished essential data or reagents; BSH, Conception and design, Drafting or revising the article; EG-W, Conception and design, Analysis and interpretation of data, Drafting or revising the article

## Additional files

### Major dataset

The following dataset was generated:

| Author(s) | Year | Dataset title | Dataset ID and/or URL | Database, license, and accessibility information |
| --- | --- | --- | --- | --- |
| Missbach C, Dweck HKM, Vogel H, Vilcinskas A, Stensmyr MC, Hansson BS, and Grosse-Wilde E | 2014 | Evolution of insect olfactory receptors - RNAseq | PRJEB5093; http://www. ebi.ac.uk/ena/data/view/ PRJEB5093 | Publicly available at the European Nucleotide Archive (http://www.ebi. ac.uk/ena/). |

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
