## [Decision Letter]

Thank you for sending your work entitled “Evolution of insect olfactory receptors” for consideration at *eLife*. Your article has been favorably evaluated by a Senior editor, Detlef Weigel, and 3 reviewers, one of whom served as a guest Reviewing editor.

The editor and the other reviewers discussed their comments before we reached this decision, and the Reviewing editor has assembled the following comments to help you prepare a revised submission.

This paper provides insights into the origins of the Odorant Receptor (OR) family of insects. By studying antennal transcriptomes of two insect species belonging to different ancient insect orders (Archaeognatha and Zygentoma) and another derived group (Phasmatodea) the authors show that the OR gene family arose later in insect diversification than has been suggested and was preceded by the Orco co-factor gene. The data do not support a previous suggestion that ORs evolved as an adaptation to terrestrialisation. In contrast to the OR family, ionotropic glutamate receptors (IRs), representing a more ancient class of chemosensory receptors, were identified and shown to be expressed in the antennal olfactory sensilla in Archaeognatha. The authors also perform anatomical and electrophysiological analyses, which provides useful novel data on the properties of the olfactory organs of these species. The paper represents a significant contribution to our understanding of insect olfaction with an interesting insight into the evolution of the OR family.

Essential revision requirements:

1) The results are largely built on the absence of OR genes in the transcriptomes of the antennae/palps of two of the lineages studied, and as acknowledged by the authors there is always uncertainty in documenting absence. These genes may be present in the genome and expressed in other chemosensory tissues not sampled here. There is a precedent for this: in the mosquito *Anopheles gambiae*, Orco and ORs have been found to be expressed in the proboscis (PMID 16938890). Given these uncertainties the text should be toned down by changing the claim that OR and Orco are lacking to OR expression not being detectable in antennae (most likely because the genes are missing from the genome).

2) The in situ images are quite hard to interpret. The important conclusion is drawn that all OSNs contain IRs, therefore supporting the idea that ORs are absent. However, the images are not very clear and poorly described in the figure legends. It is also not clear how the cell body signal is correlated with the sensillum. The authors should try to improve the presentation of these data. Fluorescent imaging in *T. domestica* gives a confusing background signal at the edge of the antenna. This artifact should be mentioned in the legend.

3) The authors should provide more detail about how OSN types were identified. The response profiles of e.g., Lys-ab5A and Lys-ab2B are similar. Why are these two types? Anatomical information ('colocalization inside the same sensillum') was also taken into account, but it is not clear how. A summary of the spatial distribution of the receptor types in the antenna would be important, if this information was used to define types. In Figure 3 some spiking rates show negative values, which indicates that the values shown represent spiking rates relative to baseline. However, there is no information about baseline activities. Since this information was also used to classify the OSN types, it should be provided.

---

## [Author Response]

*Essential revision requirements*:

*1) The results are largely built on the absence of OR genes in the transcriptomes of the antennae/palps of two of the lineages studied, and as acknowledged by the authors there is always uncertainty in documenting absence. These genes may be present in the genome and expressed in other chemosensory tissues not sampled here. There is a precedent for this: in the mosquito* Anopheles gambiae*, Orco and ORs have been found to be expressed in the proboscis (PMID 16938890). Given these uncertainties the text should be toned down by changing the claim that OR and Orco are lacking to OR expression not being detectable in antennae (most likely because the genes are missing from the genome)*.

We appreciate this comment, and of course agree. To address this problem we have included another transcriptome that is based on pooled RNA of heads and whole bodies from adults and juvenile stages. Again no ORs and Orco-like sequences were detected, supporting our initial analysis. The relevant parts of the manuscript were changed accordingly, including the addition of an accession number for the additional dataset.

*2) The* in situ *images are quite hard to interpret. The important conclusion is drawn that all OSNs contain IRs, therefore supporting the idea that ORs are absent. However, the images are not very clear and poorly described in the figure legends. It is also not clear how the cell body signal is correlated with the sensillum. The authors should try to improve the presentation of these data. Fluorescent imaging in* T. domestica *gives a confusing background signal at the edge of the antenna. This artifact should be mentioned in the legend*.

In Figure 7 part B-E have been exchanged to allow better interpretation, and we also included more panels for the same purpose. As regards Figure 9 the legend was changed to explain the basis for the mentioned correlation, and a visual marker was added to the figure itself. Additionally, we show for each antennal part the complete distribution of the different sensillum types and extended the figure legend for a better understanding. Finally we changed descriptions and interpretations of FISH in the manuscript to make statements more cautiously.

*3) The authors should provide more detail about how OSN types were identified. The response profiles of e.g., Lys-ab5A and Lys-ab2B are similar. Why are these two types? Anatomical information ('colocalization inside the same sensillum') was also taken into account, but it is not clear how. A summary of the spatial distribution of the receptor types in the antenna would be important, if this information was used to define types. In*
Figure 3
*some spiking rates show negative values, which indicates that the values shown represent spiking rates relative to baseline. However, there is no information about baseline activities. Since this information was also used to classify the OSN types, it should be provided*.

The procedure for sensillum classification has been added to the Material and methods section as requested, information on baseline activities was included into the accompanying data-source of the corresponding figure. The information of the baseline activity was not included in the figure itself, since inclusion severely impacted readability of the figure.